# Temporal and Representational Dynamics in Neural Decoding: Linear and Nonlinear Models for Position and Velocity Prediction

## Abstract

Understanding how neural activity encodes behavior remains a central challenge in systems neuroscience. Neural spike trains offer high temporal resolution and may contain information about specific motor features, but the extent of this information remains unclear. In this study, we evaluate how well motor features can be predicted from spike trains using two type of decoders: linear regression and a deep neural network. Using data from mice performing a reach-to-grab task, we compare predictions of hand position and two velocity representations. To assess whether models capture meaningful patterns rather than superficial correlations, we introduce artificial temporal lags between neural and behavioral data. This disrupts genuine associations and reveals whether decoding reflects true information content. Decoding score across lags show how behavioral information is distributed in spike trains across time and how linear and nonlinear models decode such information. This approach provides a principled framework for evaluating the behavioral relevance of neural activity.

## 1 Introduction

Understanding how the brain uses neural activity to generate behavior is a central question in systems neuroscience. Neural spike trains, which reflect time-varying activity of neuronal populations, offer rich information due to their high temporal resolution. Despite advances in recording and modeling, it remains unclear how well these signals encode behaviorally relevant variables, particularly movement.

Numerous studies have sought to decode arm movements from neural data to better understand the structured relationship between neuronal population activity and behavioral dynamics. As deep learning has rapidly emerged as a powerful tool in neural decoding (Roy et al., 2019; Livezey et al., 2019), several studies have adopted recurrent neural networks (RNNs), including long short-term memory (LSTM) architectures, to decode finger trajectories (Xie et al., 2018), support electromyographic diagnosis of neuromuscular disorders (Yoo et al., 2022) and even enable real-time robotic arm control (Du et al., 2018). Other work has leveraged temporal dependencies in electrocorticography (ECoG) signals to enable rapid and robust gesture decoding (Pan et al., 2018), demonstrated the feasibility of reconstructing three-dimensional hand velocity from noninvasive EEG (Bradberry et al., 2010), or applied deep models to integrate ECoG and video data for detecting joint movements (Wang et al., 2018). Unlike population-level signals such as ECoG or electroencephalography (EEG), which reflect spatially aggregated activity and offer limited resolution, spike trains provide millisecond-level timing and neuron-specific information. This higher specificity and temporal precision make spike trains a compelling target for deep learning-based decoding models, potentially supporting improved interpretability and physiological relevance. Accordingly, various models have been applied to decode behavior from spike trains, ranging from linear approaches such as the linear-nonlinear-Poisson (LNP) model (Paninski, 2004) and standard linear regression (Sauerbrei et al., 2020), to more advanced techniques including spiking neural networks (Kumarasinghe et al., 2021) and deep architectures based on deep canonical correlation analysis(Kim et al., 2020).

However, it remains unclear whether current decoding models are truly capturing behaviorally relevant and causally linked neural signals. Although both linear and nonlinear models have achieved

impressive performance in predicting movement trajectories, recent studies suggest that motor cortex activity can often be linearly mapped to hand trajectories (Sauerbrei et al., 2020). In particular, low-dimensional neural dynamics in the motor cortex have been shown to align closely with kinematic variables and can be linearly projected onto hand trajectories during reaching tasks (Gallego et al., 2017). Similarly, there was a report that motor cortical population activity follows rotational trajectories that correspond to specific phases of arm movement, and that these dynamics can be approximated through linear transformations of the neural state space (Russo et al., 2018). These findings support the notion that, despite the inherent complexity and nonlinearity of motor control, the relationship between motor cortical activity and certain behavioral variables can often be well approximated by linear models. Furthermore, the question of which behavioral representation—position or velocity—yields more informative or robust decoding remains unresolved.

In this study, our main contribution is :

- We systematically evaluate how temporal offsets (i.e., artificial time lags) between neural spike trains and behavioral data affect decoding performance, and compare linear and nonlinear models—including LSTM and FingerFlex—in their ability to capture behaviorally relevant patterns.

- We investigate the representational differences between predicting hand velocity and position, demonstrating that position decoding is generally more accurate and structurally stable across model architectures and temporal conditions.

- We visualize and analyze the internal representations learned by linear models, revealing that position-predictive models form smooth, temporally consistent manifolds, while velocity-predictive models form fragmented clusters—highlighting the representational simplicity and robustness of position decoding.

## 2 METHOD

Wildtype mice were attached head posts, food-restricted, and trained to reach to grab a food pellet. Neural spike trains were measured by using 384 channel four-shank silicon probes (Neuropixel (Steinmetz et al., 2021)). The primary motor cortex (M1), thalamus, striatum, and deep cerebellar nuclei activity were recorded. The recordings were spikesorted using kilosort (Pachitariu et al., 2024) and binarized with 2ms bin to generate spike trains. Recording sessions with more than 10 neurons per brain region were selected resulting in 3 sessions from one mouse. Spike trains were sliced from -250ms to +750ms from movement onset which was calculated as the time where the hand leaves the resting square box from the synchronized lateral video recording.

The sliced 1,000ms window was filtered with a Gaussian kernel ($\sigma = 10$), normalized via z-scoring. The 3D hand coordinates (Units: mm) were extracted using Animal Part Tracker (Lee et al., 2020). To make staggered data, we introduced artificial time lags in range -200 ms to + 200 ms with 40ms time bin whereas the spike data is not moved. Also to compare the decoding performance for hand position and velocity data, we applied eight-order central difference to get smoother velocity.

### 2.1 DECODING MODEL

To evaluate the capacity of neural activity in predicting behavior, we implemented and compared two distinct decoding models: a standard linear regression model and a two deep neural network architecture: Long Short-Term Memory (LSTM)(Hochreiter & Schmidhuber, 1997) and the model known as FingerFlex (Lomtev et al., 2023). The linear model assumes a direct, time-invariant mapping between the binned neural activity and behavioral output, such as hand position or velocity. Specifically, we define the linear model as follows:

$$\hat{\mathbf{y}} = \mathbf{X}\mathbf{w} + \boldsymbol{\epsilon}$$

where $\mathbf{X} \in \mathbb{R}^{D \times N}$ is the design matrix containing D time bins and N neuron channels, $\mathbf{w} \in \mathbb{R}^{N \times K}$ is the weight matrix learned during training (mapping to K behavioral output dimensions), and $\boldsymbol{\epsilon}$ is the residual error term. The model parameters $\mathbf{w}$ are obtained by minimizing the mean squared error (MSE) between the predicted and true behavioral outputs:

$$\mathbf{w}^* = \arg\min_{\mathbf{w}} \|\mathbf{X}\mathbf{w} - \mathbf{y}\|_2^2$$

which has the closed-form solution (when $\mathbf{X}^\top \mathbf{X}$ is invertible):

$$\mathbf{w}^* = (\mathbf{X}^\top \mathbf{X})^{-1} \mathbf{X}^\top \mathbf{y}$$

As a nonlinear alternative, we implemented a LSTM decoder to model the temporal dynamics in spike train data. At each time step t, the LSTM updates its hidden state $\mathbf{h}_t$ and memory cell $\mathbf{c}_t$ based on the current input $\mathbf{x}_t$ and the previous hidden state $\mathbf{h}_{t-1}$, following standard gated mechanisms:

$$(\mathbf{h}_t, \mathbf{c}_t) = \text{LSTM}(\mathbf{x}_t, \mathbf{h}_{t-1}, \mathbf{c}_{t-1})$$

The final hidden state $\mathbf{h}_t$ is projected through a fully connected layer to produce the output, and training minimizes the mean squared error (MSE) between predictions and true behavioral targets.

In addition to LSTM, we employed FingerFlex, a deep neural network originally developed for the BCI Competition IV (Miller & Schalk, 2008), where it achieved state-of-the-art performance in predicting continuous finger movements from ECoG signals. The network consists of multiple convolution layers with nonlinear activation functions, allowing it to model complex and high-order interactions across neuron channel. Authors of FingerFlex used a composite loss function combining mean squared error and cosine distance to enhance the optimization:

$$\mathcal{L}_{\text{FingerFlex}} = 0.5 \cdot \text{MSE} + 0.5 \cdot (1 - \cos(\hat{\mathbf{y}}, \mathbf{y}))$$

where $\cos(\hat{\mathbf{y}}, \mathbf{y})$ denotes the cosine similarity between predicted and true behavioral vectors.

## 3 EXPERIMENTS

To evaluate decoding performance across behavioral representations and model types, we conducted experiments using both 3D hand position and velocity data as decoding targets. The velocity signals were computed by applying an 8th-order central difference filter to the position traces. To test the effect of temporal alignment between neural activity and behavior, we introduced artificial time lags by shifting the behavioral data within a window from –200 ms to +200 ms, using a 40 ms step size. This resulted in 11 distinct lag conditions per trial.

For model validation, we employed Nested Cross-Validation (Wainer & Cawley, 2021) to ensure robust performance estimation and avoid overfitting. In this setup, the dataset was divided into k = 5 outer folds. For each outer fold, a model was trained on 4 folds and evaluated on the remaining fold to compute test performance. Within each outer fold, an additional inner k = 3 cross-validation was used solely for hyperparameter tuning or early stopping. This hierarchical scheme ensures that the test data remains completely unseen during model selection, providing an unbiased estimate of generalization performance.

As the primary evaluation metric, we used the coefficient of determination $(R^2)$ between predicted and true behavioral signals, defined as:

$$R^2 = 1 - \frac{\sum_i (\hat{y}i - y_i)^2}{\sum_i (y_i - \bar{y})^2}$$

Importantly, all $R^2$ scores—including negative values—were retained as-is when computing averages across folds and seeds, without thresholding or truncation.

All experiments were repeated across three random seeds to assess stability. For the deep neural network models, we used the Adam optimizer(Kingma & Ba, 2014) with the learning rate 0.001. Batch size is set to 32 for all methods, and the number of training epochs is 1000. Lastly all experiments were conducted on a single NVIDIA GTX 1080 GPU running on Ubuntu Linux. Model training and evaluation were implemented using PyTorch. Each deep learning model was trained for 3000 epochs.

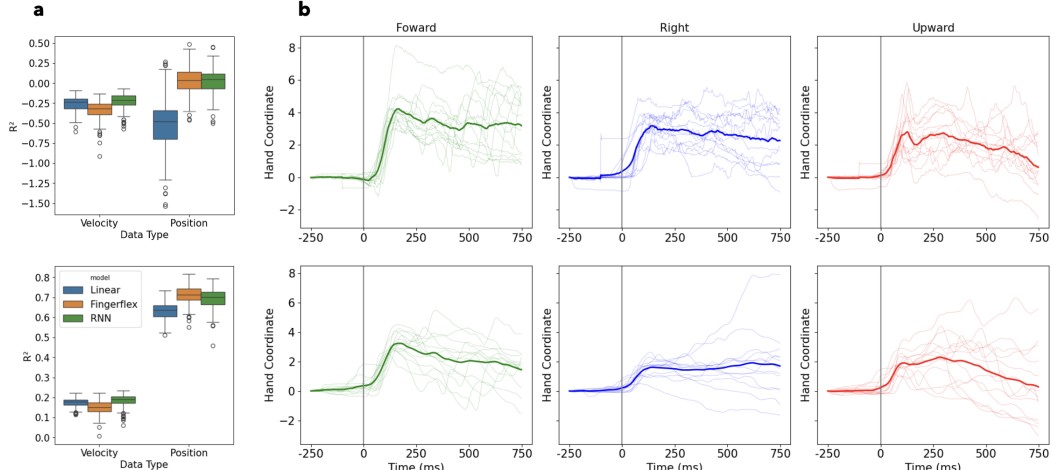

Figure 1: **(a)** Boxplots of decoding performance ($R^2$) for Linear, FingerFlex, and LSTM models across velocity and position prediction tasks. Top: $R^2$ scores computed separately for each trial and averaged. Bottom: $R^2$ scores computed from concatenated predictions across trials. **(b)** Reconstructed hand trajectories by accumulating velocity. Top: ground-truth positions aligned to movement onset. Bottom: predicted trajectories from the linear model. All results are shown for zero time lag between neural activity and behavioral data.

### 3.1 EVALUATE DECODING MODEL PERFORMANCE

We first quantified decoding performance using the coefficient of determination ($R^2$). When trial-wise $R^2$ scores were computed and subsequently averaged (Figure 1a, top), both position and velocity predictions yielded relatively low values. This is expected, as single-trial neural activity often exhibits high variability and noise, making trial-level evaluations inherently unstable. To address this, we also followed an alternative evaluation scheme commonly adopted in neural decoding studies (Sauerbrei et al., 2020), where predictions across trials were concatenated into a continuous sequence prior to computing $R^2$. Under this approach (Figure 1a, bottom), position decoding showed markedly superior performance compared to velocity decoding. These findings indicate that velocity signals are characterized by greater variance than position signals, which complicates the learning process and reduces model reliability.

### 3.2 TEMPORAL ANALYSIS

To examine how neural spike trains support decoding of different behavioral variables, we trained three models—linear regression, FingerFlex, and recurrent neural networks (LSTM)—using either velocity or position as the target. Heatmaps in Figure2 (a) and (b) show the $R^2$ scores for velocity and position decoding, respectively, across three spatial coordinates and multiple time lags.

Overall, decoding performance was consistently higher when predicting position rather than velocity, indicating that position is a more easily recoverable signal from spiking activity. Notably, the "Right" coordinate axis produced substantially lower scores across all models and behavioral variables, suggesting that this particular movement dimension is more difficult to infer from the neural population. A similar observation—that decoding performance can vary across different movement directions—has also been reported in prior studies (Sauerbrei et al., 2020). This suggests that not all behavioral components could be equally encoded in neural activity—either in quantity or representational structure.

To further examine how temporal alignment influences decoding results, we reorganized the absolute error metrics from each time-lagged model by re-aligning predictions to the movement onset. This reindexing allows visualization of prediction quality along a common temporal reference frame regardless of the lag used during training. As shown in Figure2(c) and (d), these realigned absolute error heatmaps reveal consistent spatial and temporal patterns across both linear regres-

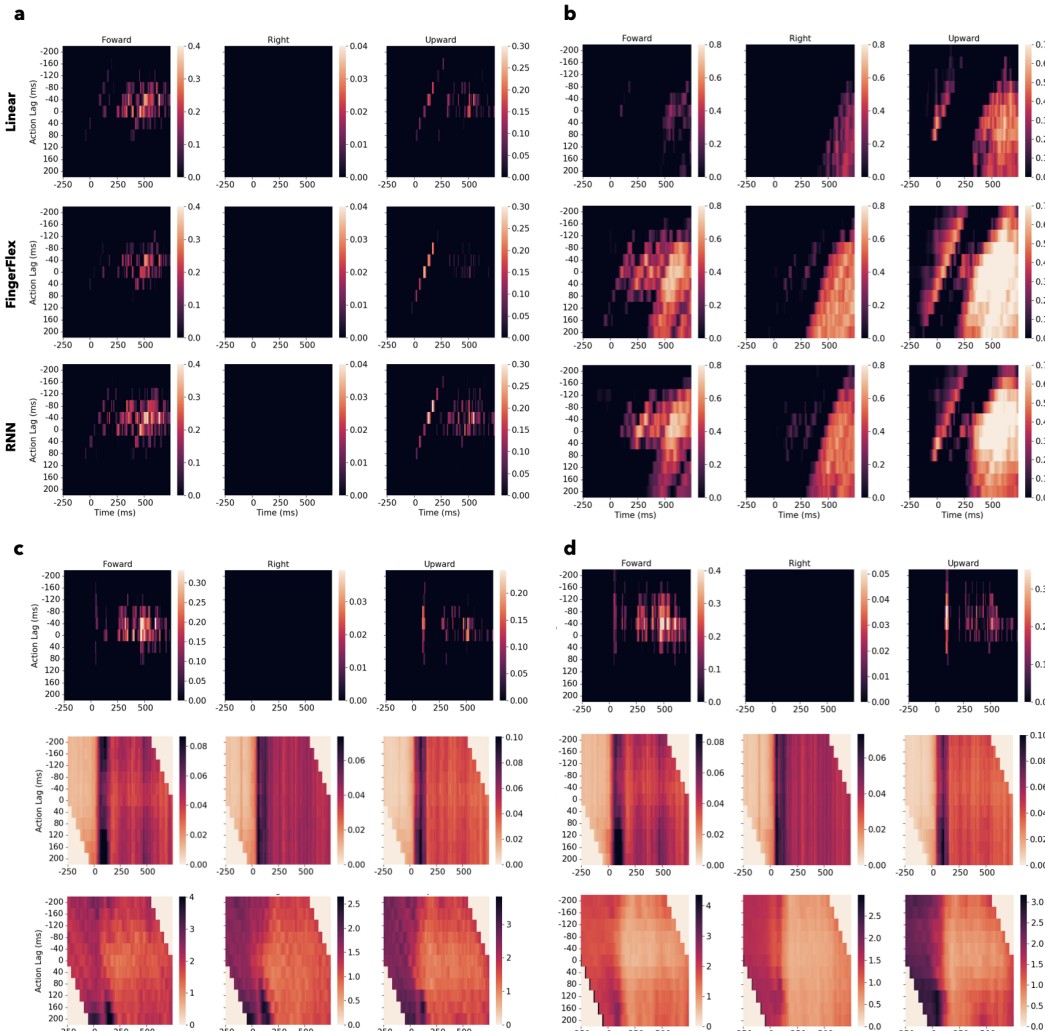

Figure 2: Heatmaps of decoding performance for different models, behavioral variables, and time lags. **(a)** $R^2$ scores for velocity decoding across three spatial coordinates (Forward, Right, Upward). **(b)** $R^2$ scores for position decoding in the same format. **(c)** Linear regression model and **(d)** LSTM heatmaps aligned to movement onset after reindexing by each time lag for : top row shows $R^2$ from velocity decoding, middle shows absolute error from velocity decoding, and bottom row shows absolute error from position decoding.

sion and LSTM models. The fact that both models—despite their structural differences in handling time—produce near-identical absolute error patterns suggests that either spike trains carry behavior-related information broadly across time, or that both models primarily act as projectors of neural trajectories, without truly capturing the underlying temporal dynamics. Given that even the time-sensitive LSTM exhibits no noticeable performance improvement over the linear model, the latter interpretation appears more likely.

For panels (c) and (d), results were averaged across the test set for each time lag. After realignment to movement onset, values outside the valid index range were filled with zero. absolute error and $R^2$ values are computed without excluding scores below zero.

### 3.3 LINEAR MODEL VISUALIZATION

To investigate how temporal delays influence the representations learned by linear decoders, we visualized the model coefficients (i.e., the learned weights) using Uniform Manifold Approximation

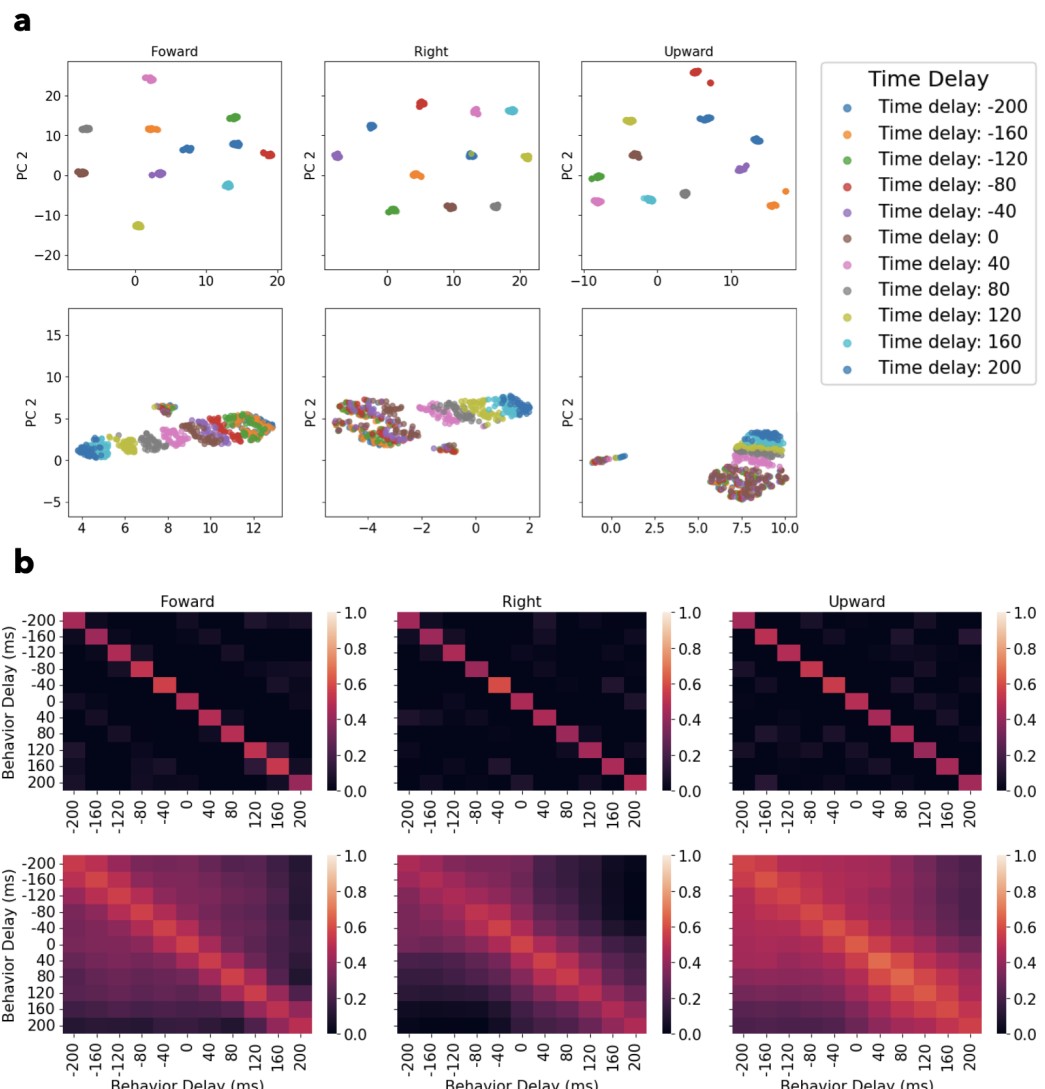

Figure 3: **(a)** UMAP projection of linear model coefficients trained with different behavior delay values. Each point represents a trained model. Top row: models trained to predict hand velocity. Bottom row: models trained to predict hand position. Colors indicate the behavior delay used during training. **(b)** Cosine similarity between learned coefficients for different delay conditions. Top: velocity prediction. Bottom: position prediction.

and Projection (UMAP)(McInnes et al., 2018), a nonlinear dimensionality reduction technique that preserves the local structure of high-dimensional data Figure 3a. Each point in the UMAP plots corresponds to a decoder trained to predict behavior at a specific time lag, with colors indicating the delay. For velocity prediction (top row), the UMAP projections show fragmented, clustered representations. Models trained at different delays form clearly separated groups, suggesting that the learned mappings are delay-specific and do not share common structure across time. This is further supported by the cosine similarity heatmaps (Figure 3b, top), which exhibit strong diagonal dominance and low off-diagonal similarity—indicating that each model captures distinct parameter spaces depending on the delay. In contrast, the position prediction models (bottom row) show a smooth progression in UMAP space, where neighboring delays yield gradually shifting representations. The corresponding cosine similarity heatmaps reveal broader similarity bands across delays, reflecting more stable and shared representations. The learned representations further reveal a fundamental difference between how models capture velocity and position. When trained on delayed

data, models predicting velocity exhibit fragmented clusters in representation space, indicating that each condition leads to a substantially different solution. In contrast, models trained on position maintain a smooth and continuous structure, even across different training conditions. Interestingly, in earlier sections, we observed that model predictions were nearly invariant to delay—regardless of whether the architecture was linear or nonlinear. This suggests that the models are not truly interpreting temporal neural dynamics, but rather projecting spike activity onto behaviorally correlated manifolds. From this perspective, the smoothness observed in position decoders indicates that position information is more structurally aligned with the spike space, making it easier to extract even with simple projection-based models. Conversely, velocity appears to require more complex, nonlinear transformations, yet still results in fragmented representations and lower decoding performance. Taken together, these findings imply that position decoding is not only more robust but also structurally easier for models to learn—regardless of whether they perform true temporal decoding or simple mapping. The continuity of representations for position supports its natural alignment with low-dimensional structure in neural activity, whereas velocity decoding poses a more challenging representational problem under the same learning conditions.

## 4 CONCLUSION

In this study, we compared the decoding performance of various models—both linear and nonlinear—in predicting hand position and velocity from spike train data. Across all conditions, hand position was consistently decoded with higher accuracy than velocity, regardless of model architecture. Although nonlinear models such as LSTM and FingerFlex theoretically offer greater representational capacity, they did not show a significant performance advantage over linear models, particularly for velocity decoding. Our results suggest that, rather than interpreting the underlying dynamics of the spike trains, both linear and nonlinear models tend to function as projection mechanisms that map neural activity onto behavioral output spaces.

Visual analyses of the learned representations further support this interpretation. Models trained on velocity exhibited fragmented and delay-specific representations, while those trained on position maintained smoother, more temporally continuous structures. This suggests that position information may be more intrinsically aligned with the structure of spike train data, making it more accessible to decoding—even in the absence of genuine temporal modeling.

### 4.1 LIMITATION

Our work has several limitations. First, the dataset was limited to a small number of recording sessions from a single mouse, which constrains the generalizability of our conclusions. Differences in neural architecture across species or task complexity may yield different encoding strategies. Second, we only explored a subset of possible model architectures—future work could benefit from examining convolutional RNNs or attention-based decoders for capturing long-term temporal dependencies. Third, our study focused on mean decoding performance and representation patterns, without explicitly quantifying statistical significance across model types and brain regions. Finally, although we explored temporal misalignment via artificial time lags, our framework does not fully disentangle causal encoding from reafferent or feedback-driven components.

### 4.2 FUTURE WORK

Future research should address several directions suggested by our findings. First, an important challenge is to design methods that can effectively handle the high variance inherent in velocity signals while maintaining robust decoding performance. Such methods may require preprocessing strategies that more effectively capture fine-grained temporal dynamics while mitigating trial-to-trial variability. Second, there is a need to develop unified models that can perform well on both position and velocity decoding, rather than being biased toward one behavioral representation. Hybrid approaches that combine recurrent dynamics with attention, graph-based connectivity, or regularization strategies may help achieve balanced performance across both tasks. Finally, validating these approaches on larger datasets, across multiple animals, and in more complex behavioral paradigms will be crucial to establishing generalizable principles of how spike trains encode both stable (position) and variable (velocity) components of movement.

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

## A    APPENDIX

You may include other additional sections here.

