# OpenReview forum: "Representational and Temporal Dynamics in Neural Decoding: Linear and Nonlinear Models for Position and Velocity Prediction"
_ICLR.cc/2026/Conference — Submitted to ICLR 2026_

### Official Review · Reviewer_Rz23 · 2025-10-20

**Soundness:** 2
**Presentation:** 2
**Contribution:** 1
**Rating:** 2
**Confidence:** 4

**Summary:**

This paper compares linear and nonlinear neural decoders (LSTM, FingerFlex) for predicting
hand position and velocity from spike trains during a reach-to-grab task in mice. The main
contribution is analyzing how artificial temporal lags between neural and behavioral data
affect decoding performance across different model architectures and behavioral variables.
The authors find that position is more decodable than velocity, and that linear models
perform comparably to nonlinear ones. While the temporal lag framework provides some utility,
the work suffers from critical limitations: severely restricted data scope (single mouse,
3 sessions), lack of novelty over prior findings, incomplete literature coverage, inadequate
baseline comparisons, and insufficient experimental depth. These limitations substantially
constrain the paper's contribution.

**Strengths:**

1. **Honest reporting of findings**: The observation that nonlinear models provide no clear
advantage over linear models is valuable and appropriately highlighted rather than being
minimized.

2. **Representation analysis via UMAP**: Figure 3 provides useful visualization of how
learned weights differ between position and velocity decoding, showing structure vs.
fragmentation patterns.

3. **Appropriate statistical practices**: Use of nested cross-validation and multiple random
seeds demonstrates care in evaluation methodology.

**Weaknesses:**

#### 1. Severely Limited Data and Generalizability

The dataset consists of **3 recording sessions from 1 mouse**, which is insufficient to
support the paper's claims or constitute a significant contribution:

- Single-animal studies cannot establish generalizable principles about neural encoding
- No assessment of across-animal variability
- No statistical significance testing across animals or brain regions

#### 2. Lack of Novelty

The core findings are not novel:

- **Linear vs. nonlinear decoding**: Already extensively studied. Sauerbrei et al. (2020),
  which the paper cites, demonstrates similar conclusions about linearity of motor cortex
  encoding.
- **Temporal lag analysis**: While useful, this is an experimental manipulation rather than
  a methodological advance.

The paper essentially replicates prior findings on new data without advancing theoretical
understanding or proposing novel methods.

#### 3. Incomplete Literature Review

Critical gaps in related work coverage:

- **No engagement with modern benchmarks** in neural decoding (post-2020 work on attention
  mechanisms, transformers, or advanced RNN variants)
- **Superficial treatment of prior studies**: Cites Gallego et al. (2017), Russo et al.
  (2018) on motor manifolds but doesn't deeply engage with their implications
- **Missing recent work**: No coverage of disentangled representations, information-theoretic
  approaches, or newer spiking neural network methods for decoding
- **Limited baseline discussion**: Does not adequately position work relative to established
  standards (Kalman filters, state-space models, particle filters)

#### 4. Inadequate Experimental Comparisons

Comparisons are extremely limited:

- Only 3 models tested (linear, LSTM, FingerFlex)
- **No classical baselines**: Kalman filters, particle filters, or other standard motor
  decoding methods
- **Multi-region data underutilized**: Records from M1, thalamus, striatum, cerebellum but
  provides no separate analysis per region
- **No ablation studies**: Which neurons matter? How many are needed?
- **No neuron/population analysis**: Feature importance, selectivity, or representational
  role not examined


### Minor Weaknesses

**Hyperparameter selection underspecified**:
   - How were LSTM/FingerFlex hyperparameters chosen?
   - Early stopping criteria?
   - What constitutes convergence?

**Questions:**

### Questions for Authors

1. **Data scope**: Why only 3 sessions from 1 mouse? Can you provide analysis on 5+ animals?
   How do results vary across animals?

2. **Brain regions**: Why are M1, thalamus, striatum, and cerebellum pooled? How do decoding
   accuracies differ per region? Which regions drive position vs. velocity decoding?

3. **Baseline comparisons**: How do results compare to Kalman filters or state-space models,
   standard baselines in motor decoding?

4. **Representation metrics**: Can you quantify representation structure beyond UMAP
   visualization (e.g., spectral properties, stability across seeds)?

5. **Velocity filtering**: How sensitive are results to the 8th-order differentiation choice?
   What if you use Savitzky-Golay or other filters?

6. **Temporal receptive fields**: How do models integrate information over time? Are they
   using history or just current activity?

7. **Statistical significance**: Were there formal tests comparing models or conditions? What
   are p-values for key claims?

---

### Official Review · Reviewer_eMbv · 2025-10-31

**Soundness:** 2
**Presentation:** 3
**Contribution:** 1
**Rating:** 0
**Confidence:** 4

**Summary:**

The paper compares linear (regression) and non-linear (LSTM) decoding techniques for relating neural spike trains and behavioral variables.

**Strengths:**

The paper is  cleanly written, the methods and experiments make sense and are well described and analyzed. The authors point out limitations (which I will still revisit below).

**Weaknesses:**

Overall, the depth, significance and scope of the experimental results does not meet the bar for ICLR. While the paper is overall well written and clear, it does not contribute much to the existing literature. The authors make very general statements from running established methods on very limited data. In this space, there is plenty of work and data/other methods available now that could have been considered.

The authors state some of the mentioned limitations themselves; I suggest to consider these and extend the work in this regard to improve the scientific depth of the paper. I am happy to engage in further discussions, but want to set expectations that it will be hard to convince me to increase my score with the present scope of the paper, and I find it close to impossible to bring this paper in a form that could be considered for acceptance. Instead, I would suggest that the authors thoroughly extend their study in terms of datasets and methods, and re-submit the work to a future venue. Please see my detailed comments below

### Introduction

- **W1.** L32-33, “it remains unclear how well these signals encode behaviorally relevant variables” — this seems like a stretch given a number of studies which are also cited later in the paper; it is unclear what the exact gap in the literature the authors want to point out. Behavior decoding is a standard method in many neuroscientific studies.
- **W2.** L35-52 has no references more recent than 2022. In the 2023–2025 a substantial amount of new methods for decoding were introduced, including more theoretically grounded ones; these should be discussed here in the light of the literature gap the authors want to close.
- **W3.** L53 it is unclear what “truly capturing behaviorally relevant and causally linked signals” means. How does the method presented here address this?

### Method

- **W4.** This is clearly written, but lacks depth. The authors run a linear regression, LSTM and FingerFlex, there is little methodological advance or connection to prior work, it is a purely empirical setup.

### Conclusions

- **W5** L 347–348: “Our results suggestion that rather than interpreting the underlying dynamics of spike trains, both linear and nonlinear models tend to function as projection mechanism that map neural activity onto behavioral output spaces” — this almost seems like a trivial statement — what does it entail to “interpret the underlying dynamics of spike trains”?

### Minor comments

- Typos in l. 152
- Equations should be numbered
- Typo in l. 158, missing space
- l. 264 typo, extra dot

**Questions:**

- **Q1.** l. 142 does the cross validation consider blocks of time, or is it randomly shuffled?
- **Q2.** Is L113 actually how the regression analysis was done, or was there regularization?
- **Q3.** What motivates the choice of these particular models, vs. other established models for decoding? This might be related to the issue outlined above, no methods past 2022 are discussed in the literature review.

---

### Official Review · Reviewer_TdLe · 2025-11-01

**Soundness:** 2
**Presentation:** 2
**Contribution:** 4
**Rating:** 2
**Confidence:** 2

**Summary:**

This paper uses both linear and nonlinear models training on neural activity data for position and velocity prediction. To assess whether models capture meaningful patterns rather than superficial correlations, they introduce artificial temporal lags between neural and behavioral data.

**Strengths:**

The paper evaluated how temporal offsets (i.e., artificial time lags) between neural spike trains and behavioral data affect decoding performance, and compare linear and nonlinear modelsin their ability to capture behaviorally relevant patterns.

The paper investigate the representational differences between predicting hand velocity and position, demonstrating that position decoding is generally more accurate and structurally stable across model architectures and temporal conditions.

The paper visualize and analyze the internal representations learned by linear models, revealing that position-predictive models form smooth, temporally consistent manifolds, while velocity- predictive models form fragmented clusters—highlighting the representational simplicity and robustness of position decoding.

**Weaknesses:**

The paper uses simple linear regression and simple models including LSTM and FingerFlex for the comparison, which is too simple and trivial. There's not too much sense for that.

The paper investigated the time lags with neural spike trains only by showing their $R^2$ score in heatmap.

**Questions:**

On page 4, Figure 1 (a), can you explain why you use box plot for the $R^2$ values?

On page 4, Figure 1 (b), can you explain what the light lines and dark line in each pair mean? How can you tell the corresponding ground truth and prediction in each train? How did you do the prediction? What is given for the generation of a train?

---

### Official Review · Reviewer_X7MY · 2025-11-01

**Soundness:** 1
**Presentation:** 1
**Contribution:** 1
**Rating:** 0
**Confidence:** 5

**Summary:**

The paper compares how neural spike recordings encode different movement variables (position, velocity) with different models (linear regression, LSTM, FingerFlex).  The authors introduce artificial temporal lags between behavior and neural recording to probe the causal relationship between the lag and decoding performance. They perform UMAP to compare the internal representation of position, velocity trained linear models.

**Strengths:**

The authors collected new neural and behavioral data from mice. They compared different behavioral decoding variables and model types.

**Weaknesses:**

1. The paper lacks genuine insights drawn from different experimental results. The results are presented in fragmented manners and the authors did not connect those observations to make a coherent conclusion. Furthermore, many results seem to have already been shown in different papers, as the authors already cited.

2. Many figures are lacking proper panel titles and axis labels.

3. There is no detail about hyperparameter choice - since the dataset is small neural data, controlling for overfitting is important, but I don’t see any justification with the validation loss or hyperparameters search.

**Questions:**

I do not think the paper is ready for publication. I recommend that the author work on what questions they are exactly asking, what experiments are needed and more rigorous experiments to draw a meaningful conclusion.

---

### Meta-Review · Area_Chair_AGtL · 2026-01-03

**Summary:**

This paper studies how neural spike trains encode behavioral variables by comparing linear and nonlinear decoders for predicting hand position and velocity in a mouse reach-to-grab task. A key methodological element is the introduction of artificial temporal lags between neural activity and behavior to probe whether decoding performance reflects meaningful temporal structure rather than superficial correlations. The authors report that position is more reliably decodable than velocity and that simple linear models perform comparably to nonlinear models. While the experiments are generally sound and clearly described, the reviewers consistently raised concerns regarding the paper’s limited novelty, insufficient experimental scope, and lack of deeper insight beyond well-established findings in the neural decoding literature. Overall, the work was viewed as a replication-style study on a small dataset that does not meet the contribution bar for ICLR.

**Reviewer Concerns:**

There is no rebuttal.

**Reviewer Scores:**

Based on the discussion and the nature of the rebuttal, it is unlikely that reviewer scores would have changed materially:

Reviewer X7MY: Strong reject (0). The reviewer’s primary concerns about lack of insight, coherence, and experimental rigor were not substantially addressed. Score would remain unchanged.

Reviewer TdLe: Reject (2). While this reviewer found some aspects interesting, their concerns about triviality of models and limited depth were not resolved. A score increase is unlikely.

Reviewer eMbv: Strong reject (0). The reviewer explicitly stated that it would be close to impossible to bring the paper to acceptance without a major extension in scope and methods. Score would remain unchanged.

Reviewer Rz23: Reject (2). The reviewer’s detailed concerns about data limitations, novelty, and missing baselines remain outstanding. No score change is expected.

---

### Decision · Program_Chairs · 2026-01-26

Reject